# Temporal Mental Health Dynamics on Social Media

**Tom Tabak**[1]         **Matthew Purver** [1] [2]

[1]School of Electronic Engineering and Computer Science
Queen Mary University of London
[2]Department of Knowledge Technologies, Jožef Stefan Institute
tabaktom360@gmail.com         m.purver@qmul.ac.uk

## Abstract

We describe a set of experiments for building a temporal mental health dynamics system. We utilise a pre-existing methodology for distant-supervision of mental health data mining from social media platforms and deploy the system during the global COVID-19 pandemic as a case study. Despite the challenging nature of the task, we produce encouraging results, both explicit to the global pandemic and implicit to a global phenomenon, Christmas Depression, supported by the literature. We propose a methodology for providing insight into temporal mental health dynamics to be utilised for strategic decision-making.

## 1   Introduction

Mental health issues pose a significant threat to the general population. Quantifiable data sources pertaining to mental health are scarce in comparison to physical health data (Coppersmith et al. 2014). This scarcity contributes to the complexity of development of reliable diagnoses and effective treatment of mental health issues as is the norm in physical health (Righetti-Veltema et al. 1998). The scarcity is partially due to complexity and variation in underlying causes of mental illness. Furthermore, the traditional method for gathering population-level mental health data, behavioral surveys, is costly and often delayed (De Choudhury, Counts & Horvitz 2013b).

Whilst widespread adoption and engagement in social media platforms has provided researchers with a plentiful data source for a variety of tasks, including mental health diagnosis; it has not, yet, yielded a concrete solution to mental health diagnosis (Ayers et al. 2014). Conducting mental health diagnosis tasks on social media data presents its own set of challenges: The users' option of conveying a particular public persona posts that may not be genuine; sampling from a sub-population that is either technologically savvy, which may lend to a generational bias, or those that can afford the financial cost of the technology, which may lead to a demographic bias. However, the richness and diversity of the available data's content make it an attractive data source. Quantifiable data from social media platforms is by nature social and crucially (in the context of our cases study) virtual.

Quantifiable social media data enables researchers to develop methodologies for distant mental health diagnosis and analyse different mental illnesses (De Choudhury, Counts & Horvitz 2013a). Distant detection and analysis enables researchers to monitor relationships of temporal mental health dynamics to adverse conditions such as war, economic crisis or a pandemic such as the Coronavirus (COVID-19) pandemic.

COVID-19, a novel virus, proved to be fatal in many cases during the global pandemic that started in 2019. Governments reacted to the pandemic by placing measures restricting the movement of people on and within their borders in an attempt to slow the spread of the virus. The restrictions came in the form of many consecutive temporary policies that varied across countries in their execution. We focus on arguably the most disruptive measure: The National Lockdown. This required individuals, other than essential workers (e.g. healthcare professionals) to remain in their own homes. The lockdown enforcement varied across countries but the premise was that individuals were only permitted to leave their homes briefly for essential shopping (food and medicine). This policy had far reaching social and economic impacts: growing concern towards individuals' own and their families' health, economic well-being and financial uncertainty as certain industries (such as hospitality, retail and travel) suspended operations. As a result, many individuals became redundant and unemployed which constrained their financial re-

sources as well as being confined to their homes, resulted in excess leisure time. These experiences along with the uncertainty of the measures' duration reflected a unique period where the general public would be experiencing a similar *stressful* and *anxious* period, which are both feelings associated with clinical depression (Rickels & Schweizer 1993, Hecht et al. 1989).

In this paper, we investigate the task of detecting whether a user is diagnosis-worthy over a given period of time and explore what might this appropriate time period be. We investigate the role of balance of classes in datsets by experimenting with a variety of training regimes. Finally, we examine the temporal mental health dynamics in relations to the respective national lockdowns and investigate how these temporal mental health dynamics varied across countries highly-disrupted by the pandemic. Our main contributions in this paper are: 1) We demonstrate an improvement in mental health detection performance with increasingly enriched sample representations. 2) We highlight the importance of the balance in classes of the training dataset whilst remaining aware of an approximated expected balance of classes in the unsupervised (test) dataset. 3) We analyse empirically proven relationships between populations' temporal mental health dynamics and respective national lockdowns that can be used for strategic decision-making purposes.

## 2 Related research

### 2.1 Natural Language Processing for Mental Health Detection

Unlike physical health conditions that often show physical symptoms, mental health is often reflected by more subtle symptoms (De Choudhury, Counts & Horvitz 2013*a*, Chung & Pennebaker 2007). This yielded a body of work that focused on linguistic analysis of lexical and semantic uses in speech, such as diagnosing a patient with depression and paranoia (Oxman et al. 1982). Furthermore, an examination of college students' essays, found an increased use of negative emotional lexical content in the group of students that had high scores on depression scales (Rude et al. 2004). Such findings confirmed that language can be an indicator of an individual's psychological state (Bucci & Freedman 1981) which lead to the development of Linguistic Enquiry and Word Count (LIWC) software (Pennebaker et al. 2003, Tauszik & Pennebaker

2010) which allows users to evaluate texts based on word counts in a variety of categories. More recent and larger scale computational linguistics have been applied in conversational counselling by utilising data from an SMS service where vulnerable users can engage in therapeutic discussion with counsellors (Althoff et al. 2016). For a more in-depth review of uses of natural language processing (NLP) techniques applied in mental health the reader is referred to Trotzek et al. (2018).

### 2.2 Social Media as a Platform for Mental Health Monitoring

The widespread engagement in social media platforms by users coupled with the availability of platforms' data enables researchers to extract population-level health information that make it possible to track diseases, medications and symptoms (Paul & Dredze 2011). The use of social media data is attractive to researchers not only due to its vast domain coverage but also due to the cheap methodologies by which data can be collected in comparison to previously available methodologies (Coppersmith et al. 2014). A plethora of mental health monitoring literature have utilised this cheap and efficient data mining methodologies from a variety of social media platforms such as: Reddit (Losada & Crestani 2016), Facebook (Guntuku et al. 2017) and Twitter (De Choudhury, Gamon, Counts, & Horvitz 2013).

Twitter user's engagement in the popular social media platform give way for the creation of social patterns that can be analysed by researchers, making this platform a widely used data source for data mining. Additionally, the customisable parameters querying available in the Application Programmable Interface (API) allows researchers to monitor specific populations and/or domains (De Choudhury, Counts & Horvitz 2013*b*).

### 2.3 Mental Health Monitoring During COVID-19 Pandemic

In the context of the COVID-19 pandemic, we found a handful of projects with similar intentions as our own, to monitor depression during the pandemic. Li et al. (2020) gather large scale, pandemic-related twitter data and infers depression based on emotional characteristics and sentiment analysis of tweets. Zhou et al. (2020) focus on detecting community level depression in Australia during the pandemic. They use the distant-supervision methodologies of Shen et al. (2017) to gather a

balanced dataset, they utilise the methodology of Coppersmith et al. (2014) to model the rates of depression and observing the relationship with the number of COVID-19 infections in the community. Our work differs from this in three main areas: 1) We investigate the implication of different sample representations to provide more context to our classifier. 2) We retain an imbalance in our development dataset. 3) We investigate European countries (France, Germany, Italy, Spain and the United Kingdom) that experienced a relatively high number of COVID-19 infections.

## 3 Diagnosis Classifier Experiments

We describe the data mining methodology used to build a distantly supervised dataset and the classifier experiments conducted on this dataset.

### 3.1 Data

To conduct the proposed experiments, we construct a distantly supervised development dataset for each country, to be used in training and validation of the classifier. The data mining methods follow the novel distant-supervision methodology proposed in Coppersmith et al. (2014) as it is relatively cheap but also well-structured for clinical experiments.

We follow the wide-accepted methodology proposed by Watson (1768) where diagnosed (`Diagnosed`) and non-diagnosed (`Control`), groups are created. In this paper we will only be exploring depression as a mental health condition, accordingly we will have a single `Diagnosed` group for each country's development dataset. However, if multiple mental issues were to be explored, then the same number of different `Diagnosed` groups would be required for each country's dataset.

### 3.1.1 `Diagnosed` Group

We gather 200 public tweets with a geolocation inside the country of interest, posted during a two-week period (1 July 2019 - 15 July 2019). As we are searching for a *depression* `Diagnosed` tweets, this two-week period needs to be chosen strategically, as we want to capture users that have been diagnosed with depression rather than seasonal affect disorder (SAD), a separate albeit a condition with similar symptoms. Tweets collected via Twitter's API[1], were retrieved based on lexical content

---

[1]Twitter API: `https://developer.twitter.com/en/docs`

indicating that the user has history/is currently dealing with a clinical case, e.g. "I was diagnosed with depression", rather than expressing depression in a colloquial context. Human annotators were then instructed to remove tweets that are perceived to have made a non-genuine statement regarding the users' own diagnosis, most of these were referring to a third party. Examples of genuine and non-genuine tweets encountered can be seen in *Table 1*.

Table 1: Annotation Example

| Diagnosis indication | Example tweet |
|---|---|
| Genuine | "I was diagnosed with severe depression and went through the works of treatment for it." |
| Non-genuine | "It's official. My guinea pig has been diagnosed with depression" |

We then collect all (up to 5,000 most recent) tweets made public by the remaining users between the start of 2015 and October 2019. Further filtering includes removal of all users with less than 20 tweets during this period or those whose tweets do not meet our major language of instruction benchmark. This benchmark requires 70% of the tweets collected to be written in the major language of instruction of the country of interest (i.e. United Kingdom is English, Italy is Italian etc.). Following this filtering process and some preprocessing on the tweet level, which includes medial capital splitting, mention white-space removal (i.e. if another user was mentioned this will be shown as a unique *mention* token), the same has been done with URLs, all uppercase and non-emoticon related punctuation were removed.

### 3.1.2 `Control` Group

We gather 10,000 public tweets with a geolocation in the country of interest, posted during the same two-week period as `Diagnosed` in 2019 and remove any tweets made by `Diagnosed` users. We then follow a similar process to that of `Diagnosed` collection methodology by collecting up to 5,000 most recent tweets for each user from the period mentioned above.

As can be seen in *Table 2*, we construct imbalanced datasets. World Health Organisation (WHO) claim 264 million people suffer from depression worldwide[2]. Whilst, at the time of writing, the

---

[2]World Health Organization, "Depression," 2020, [Online]. Available: `https://www.who.int/news-room/fact-sheets/detail/depression`[Accessed: 26 July 2020]

Table 2: Composition of Development Datasets

| Country | Group | No. Users | No. Tweets |
|---------|-------|-----------|------------|
| France | `Diagnosed` | 57 | $190,447$ |
| | `Control` | $1,041$ | $2,861,580$ |
| Germany | `Diagnosed` | 53 | $160,864$ |
| | `Control` | $1,138$ | $2,802,959$ |
| Italy | `Diagnosed` | 38 | $132,743$ |
| | `Control` | $1,051$ | $2,514,483$ |
| Spain | `Diagnosed` | 53 | $107,833$ |
| | `Control` | $1,013$ | $2,564,966$ |
| U.K. | `Diagnosed` | 98 | $289,624$ |
| | `Control` | $1,365$ | $3,319,201$ |

global population stands at approximately 7.8 billion[3]. This would suggest that 1 in 30 individuals suffer from depression. However, these figures are approximations. Therefore, the extent to which our datasets are imbalanced is not an attempt to create datasets that are representative of the expected balance of classes, as these are unverifiable. Nevertheless, our datasets present ratios of `Control:Diagnosed` samples between 23.78:1 and 11.46:1, which came about from the data mining methods previously described. We accept these ratios to retain imbalanced datasets in a similar order of magnitude as the expected balance whilst achieving reasonable classifier performance.

### 3.1.3 Caveats

We inherit the limitations of the distant-supervision approach of Coppersmith et al. (2014):

1. When sampling a population we always run the risk of only capturing a subpopulation of `Control` or `Diagnosed` that is not fully representative of the population, especially considering that the `Diagnosed` group are identified based a single affirming tweet about an intimate subject – this attribute may not generalise well to the entire population.

2. We supervise all tweets of a unique user based on a single affirming tweet. Hence, this may result in different tweets with identical, or similar, meaning representations being assigned different labels.

3. We do not implement a verification of the method used to identify users in the

[3]Worldometer. 2020. Worldometer - Real Time World Statistics. [online] Available at: https://www.worldometers.info [Accessed 19 August 2020].

`Diagnosed` group but rather rely on the social stigma around mental illness whereas it could be regarded as unusual for a user to tweet about a diagnosis of a mental health illness that is fictitious.

4. `Control` is likely contaminated with users that are diagnosed with a variety of conditions, perhaps mental health related, whether they explicitly mention this or not. We have made no attempt to remove such users.

5. Twitter users may not be entirely representative sample of the population.

### 3.2 Methodology

We describe the experiments conducted in classifier training of depression diagnosis. The trained classifier is deployed in *Section 4* for classifying samples from an unsupervised experiment dataset which is then used in analysing temporal mental health dynamics.

### 3.2.1 Sample Representation

We investigate the most appropriate sample representation of our distantly supervised dataset. We are posed with these considerations:

1. Symptoms' temporal dependencies: as the tweets gathered come from a variety of days, weeks, months and even years, symptoms may only be present in specific time-dependant samples. However, when represented by overwhelming tweet-enriched samples the classifier performance is traded-off with retaining the symptoms' temporal dependencies.

2. As our final task will be to monitor and analyse the temporal mental health dynamics, we are interested in modelling the rate of depression as fine-grained as possible.

Therefore, the ability to accurately identify `Diagnosed` samples and correctly discriminate between `Control` and `Diagnosed` with the least tweet-enriched samples will be vital in modelling a fine-grained rate of depression in the deployment stage of the final task where conclusions could be drawn in the context of the national lockdowns. The sample representations we examined:

- *Individual* – each sample constitutes of a single tweet.

- *User day* - each sample constitutes of all tweets by a unique user during a given day.

- *User week* – each sample constitutes of all tweets by a unique user during a given week.
- *All user* - each sample constitutes of all tweets collected from a unique user.

We examine the performance of a benchmark, Support Vector Machine (SVM) with a linear kernel function (Peng et al. 2019), on the different sample representations datasets where the benchmark classifier inputs are sparse many-hot encoding representations of the samples' lexical content. As we are working with imbalanced datasets we need to think about the metrics we use to assess the classifiers' performance. We will be assessing class specific Precision (P) and Recall (R) as well as Macro F1 score. By having a more class-specific breakdown of the classifiers' performance we can better understand the strengths and limitations of our classifiers and hence make a more informed decision when choosing the highest performing classifier.

Table 3: SVM Performance on Varying Sample Representations of U.K. Development Datasets

| Sample Representation | Control | | | Diagnosed | | | Macro F1 |
|---|---|---|---|---|---|---|---|
| | P | R | F1 | P | R | F1 | |
| *Individual** | 0.92 | 0.99 | 0.95 | 0.27 | 0.05 | 0.08 | 0.52 |
| *User day* | 0.74 | 0.96 | 0.84 | 0.36 | 0.06 | 0.1 | 0.52 |
| *User week* | 0.92 | 0.97 | 0.94 | 0.26 | 0.11 | 0.15 | 0.55 |
| *All user* | 0.94 | 0.99 | 0.96 | 0.5 | 0.14 | 0.22 | 0.59 |

The results in *Table 3* suggest that our benchmark classifier improved in identifying `Diagnosed`, with increasingly tweet-enriched, samples. However, the *User day* sample representations shows a decrease in performance when compared with the F1 scores of both *Individual* and *User week* sample representations. Barring this decrease, we can say that we are able to achieve improved performance when using increasingly tweet-enriched samples. However, our final task is bias towards the two fine-grained sample representations, *Individual* and *User day*. The benchmark classifier achieves superior performance on the *Individual* sample representation, we will adopt this representation in further experiments, as denoted by the asterisk in *Table 3*.

### 3.2.2 Classifier Experiments on U.K. Development Dataset

We must now build and train a classifier architecture that best discriminates between our two classes. Classifier architectures included in our experimentation: $SVM$: SVM as used in *Section 3.2.1*. This classifier will serve as our benchmark; $AVEPL_{EFC}$ [4]: Average pooling layer; $CNN\text{-}MXPL_{EFC}$: CNN[5] and a Max-pooling layer; $BILSTM_{EFC}$: Bi-directional LSTM (Hochreiter & Schmidhuber 1997); $CNN\text{-}BILSTM_{EFC}$: CNN and Bi-directional LSTM; $CNN\text{-}ATT$: CNN, Attention (Vaswani et al. 2017) and Average pooling layers; $BILSTM\text{-}SELFA$: Bi-directional LSTM and Self-attention layer; and $BERT$: Pretrained $BERT_{Base}$ [6] fine-tuned on our dataset (Devlin et al. 2018). All classifiers use an Adam optimiser (Kingma & Ba 2015) and were trained for a single epoch on a training:validation split of 4:1 with weighting the `Diagnosed` samples as 5 times more valuable than those of `Control`. Training on a single epoch was chosen in line with our theme of quick development (also conveyed in the distant-supervision), we argue that performance could be improved by further training. The sample weighting factor was chosen following empirical evidence showing that the chosen factor yielded similar `Diagnosed` Precision and Recall measures.

Table 4: Classifiers' Performance on U.K. Development Dataset

| Classifier | Control | | | Diagnosed | | | Macro F1 |
|---|---|---|---|---|---|---|---|
| | P | R | F1 | P | R | F1 | |
| $SVM$ | 0.92 | 0.9 | 0.91 | 0.27 | 0.05 | 0.08 | 0.52 |
| $AVEPL$ | 0.94 | 0.95 | 0.94 | 0.33 | 0.26 | 0.29 | 0.62 |
| $CNN\text{-}MXPL$ | 0.94 | 0.95 | 0.94 | 0.31 | 0.25 | 0.28 | 0.61 |
| $BILSTM$ | 0.94 | 0.95 | 0.94 | 0.3 | 0.33 | 0.31 | 0.63 |
| $CNN\text{-}BILSTM$ | 0.93 | 0.97 | 0.95 | 0.3 | 0.15 | 0.2 | 0.57 |
| $CNN\text{-}ATT$ | 0.94 | 0.94 | 0.94 | 0.31 | 0.3 | 0.3 | 0.62 |
| $BILSTM\text{-}SELFA*$ | 0.94 | 0.94 | 0.94 | 0.32 | 0.31 | 0.31 | 0.63 |
| $BERT$ | 0.94 | 0.66 | 0.78 | 0.12 | 0.52 | 0.2 | 0.47 |

*Table 4* shows all classifiers achieve significantly higher performance on `Control` than the `Diagnosed`. The first observation is with respect to the poor performance of $BERT$ with a *Macro F1* of 0.47, lower than our benchmark $SVM$ classifier. We argue that this poor performance can be attributed to the extensively trained word embeddings of the $BERT$ classifier remaining under-

---

[4] All uses of $X_{EFC}$ indicate a learned embedding layer, after the input, and 3 Fully Connected layers, with Rectified Linear (ReLU), directly prior to the output layer.

[5] All CNNs are unigram-level with 1 $filter$ and $kernel\ size$ of 1.

[6] Exact pretrained $BERT_{Base}$ version implementation available here: https://tfhub.dev/google/bert_uncased_L-12_H-768_A-12/1

utilised due to our classifier's input, which contains many spelling errors. Admittedly, this hypothesis is mere conjecture and we leave this topic for future work. As we are trying to correctly detect `Diagnosed` samples and discriminate between the two classes, we prioritise the `Diagnosed` *Precision* and *Macro F1* score metrics. Based on these 2 chosen metrics to guide our classifier selection process 3 candidates emerge: $AVEPL$, $BILSTM$ and $BILSTM$-$SELFA$ achieving {`Diagnosed` *Precision*, *Macro F1*} scores of: {0.33, 0.62}; {0.3, 0.63} and {0.32, 0.63} respectively. Whilst the performance of these classifiers is similar, $BILSTM$-$SELFA$ is the highest performance combination of the desired metrics (indicated by the asterisk) and as such we will be adopting this classifier in further experiments.

### 3.2.3 Dataset Balance Experiment

In this section we investigate the distribution of our datasets in training and validation of our classifier. By conducting this experiment we intend to gather an in-depth understanding of our task from a linguistic standpoint. We train and validate the classifier on datasets with varying balances to investigate the role of our imbalanced dataset in the depression diagnosis task. This experiment analyses the performance of the $BILSTM$-$SELFA$ classifier on a number of different training regimes:

- *Balanced*: a dataset containing all `Diagnosed` samples and downsampling from `Control`.

- *Imbalanced*: a dataset of the development dataset's distribution (See *Table 2*).

Furthermore, we explore the effects of sample weighting by weighting `Diagnosed` samples as 5 times more valuable than the `Control` samples as mentioned in *Section 3.2.2*. The performance of the $BILSTM$-$SELFA$ classifier on the different training regimes can be seen in *Table 5*.

Table 5: $BILSTM$-$SELFA$ Performance on Varying Training Regimes

| Training | Validation | Sample Weighting | Control | | | Diagnosed | | | Macro |
|---|---|---|---|---|---|---|---|---|---|
| | | | P | R | F1 | P | R | F1 | F1 |
| *Balanced* | *Balanced* | None | 0.69 | 0.69 | 0.69 | 0.69 | 0.69 | 0.69 | 0.69 |
| *Imbalanced* | *Imbalanced* | None | 0.93 | 1 | 0.96 | 0.72 | 0.11 | 0.19 | 0.58 |
| *Imbalanced* | *Imbalanced* | Weighted | 0.94 | 0.94 | 0.94 | 0.32 | 0.31 | 0.31 | 0.63 |
| *Balanced* | *Imbalanced* | None | 0.95 | 0.63 | 0.76 | 0.14 | 0.66 | 0.23 | 0.49 |
| *Balanced* | *Imbalanced* | Weighted | 0.99 | 0.13 | 0.23 | 0.09 | 0.98 | 0.16 | 0.2 |
| *Imbalanced* | *Balanced* | None | 0.53 | 1 | 0.69 | 0.98 | 0.11 | 0.2 | 0.45 |
| *Imbalanced* | *Balanced* | Weighted | 0.58 | 0.96 | 0.72 | 0.88 | 0.29 | 0.44 | 0.58 |

The *Balanced-Balanced* training regime achieves an encouraging Precision-Recall trade-off, for both classes, as well as the Macro F1 score. This shows that the problem is reasonably linguistically achievable, when the imbalance challenge is removed. The *Imbalanced-Imbalanced* training regime shows that adjusting the sample weighting is a successful measure that we can implement to adjust the Precision-Recall trade-off in our class of interest (`Diagnosed`). Our classifier performs significantly worse in the *Balanced-Imbalanced* regime when compared to the performance on the *Imbalanced-Imbalanced* regime, this performance is reduced by the introduction of sample weighting. Therefore, when training on a *Balanced* dataset our classifier is less robust to an *Imbalanced* validation dataset. Finally, whilst our classifier experiences a significant improvement in performance on the *Imbalanced-Balanced* training regime when sample weighting is introduced due to our final depression diagnosis task in which we expect an *Imbalanced* dataset (see *Section 3.1.2*) the training regimes implementing *Balanced* validation datasets are not suitable approximations of our classifier's depression diagnosis performance. Therefore, the *Imbalanced* training, with suitable sample weighting, yields more desirable and robust depression diagnosis performance as it is exposed to a broader range of data examples in training (i.e. no sub-sampling).

### 3.3 Results

We train separate $BILSTM$-$SELFA$ classifiers on each of the respective countries' imbalanced development datasets following the *Individual* sample representation (see *Table 7 Appendix A.1*). We observe that the $BILSTM$-$SELFA$ architecture achieved similar performance on the remaining countries' datasets. Whilst the $BILSTM$-$SELFA$ classifier architecture achieved the highest performance of all our classifier architectures, a combination of 0.32 `Diagnosed` *Precision* and 0.63 *Macro F1* leaves much to be desired. As such, we perform an error analysis and examine the significance of the results.

#### 3.3.1 Error Analysis

*Table 6* shows the input samples, *Text*, the *Prediction type* as well as the *Sigmoid Output* which is the output layer of the classifier and is responsible for the final classification of the samples. The *Sigmoid Output* is normalised in the range of

$[0, 1] \in \mathbb{R}$, where an output of 0.5 represent the decision boundary, and is interpreted as complete uncertainty with regards to the sample's classification. A *Sigmoid Output* of 1 is complete certainty that the sample should be classified as `Diagnosed` and 0 is complete certainty in `Control`.

Table 6: Classification Examples for Error Analysis

| Prediction Type | Text | Sigmoid Output |
|---|---|---|
| True Positive | "hi davenport handmade is a small one man business i make handmade wooden bowls pens jewellery boxes and other wooden items in a workshop that i built myself it started as a way of overcoming depression and has taken over my life" | 0.999 |
| False Positive | "im too depressed lol" | 0.507 |
| False Negative | "i miss you too man its actually depressing me" | 0.19 |
| True Negative | "half term kids camps are up on wandsworth common with a dedicated kids football camp" | 0.001 |

The true positive example mentions having "overcoming depression" which implies that the user has recovered from depression, as one overcomes other health issues. The *Sigmoid Output* is 0.999 which is extremely high certainty by the classifier that this is a `Diagnosed`. Whilst, the true negative is unrelated to depression nor its underlying symptoms, as such it is classified as part of `Control` with a *Sigmoid Output* of 0.001. However, the *Texts* of the misclassified samples are similar. Both use words stemming from the word 'depress' in colloquial contexts, with no indication of clinical appropriations of depression. The *Sigmoid Outputs* of these samples are less polarised than those correctly classified, the *Sigmoid Output* of the false positive sample is marginally misclassified. However, these misclassified samples reflect the complexity of the task.

### 3.3.2 Significance of Results

We perform a $\chi^2$ significance test to investigate the significance of our classifiers' results. Our null hypothesis, $H_0$, states that both sets of data, our classifiers' predictions ($\mathcal{D}_P$) and the distribution it is being tested against ($\mathcal{D}_T$), have been drawn from the same distribution ($\mathcal{D}$).

$$H_0 : \mathcal{D}_P \cap \mathcal{D}_T \subseteq \mathcal{D} \qquad (1)$$

We compare the distribution of the classifiers' predictions against a random uniformly distributed set (Uniform) and against a random distributed set following the distribution of the development datasets (Weighted). All classifier results in *Table 7* are statistically significant from the random baselines, according to the $\chi^2$ significance test (see *Table 8* in *Appendix A.2*). Therefore, we reject $H_0$ and conclude that the classifiers' predictions and those

of the respective randomly distributed benchmarks have not been drawn from the same distribution.

## 4  Monitoring and Analysis

We prepare the unsupervised dataset and deploy the previously trained $BILSTM\text{-}SELFA$ classifier to annotate this dataset. We analyse the relationships between the temporal mental health dynamics and respective national lockdowns.

### 4.1  Data

We discuss the procedure for constructing the unsupervised experiment dataset, to be used for monitoring the temporal mental health dynamics.

#### 4.1.1  Experiment Dataset

We gather tweets made public by users during the first two weeks of 2020 with a geolocation within the country of interest. We then follow the same methodology as outlined in *Section 3.1*, for the period of 1 December 2019 until 15 May 2020. The composition of these experiment datasets can be seen (*Table 9* in *Appendix A.3*) along with key dates. The key dates specified observe the official date announcements of the commencement of and of the first step towards easing of national lockdowns, rather than the first official data implementing these measures as we anticipate that the announcements would provoke users to express their opinion more than the implementation of the measures. We acknowledge caveats to the methodology with relations to the respective national lockdowns:

1. The activity-level of users whose lifestyles have been highly disrupted by the national lockdowns may be overstated during this period, due to increased leisure time.

2. The language filtering component excludes certain users of the population, such as stranded tourists/expatriates, that use a non-majority languages. Such samples may contain a bias towards a higher rate of depression.

### 4.2  Methodology

To monitor and analyse temporal mental health dynamics we must deploy our trained $BILSTM\text{-}SELFA$ on the respective countries' experiment datasets. Once we have the classifier's predictions, we calculate the rate of depression at any given day, $R_t$:

$$R_t = \frac{\sum_{i=1}^{N_t} \Phi(x_i)}{N_t} \qquad (2)$$

Where $\Phi$ represents our trained classifier, $x_i$ is the input, $N_t$ is the total number of samples on day $t$. The output of the classifier, $\Phi(x_i)$ takes the form $[0, 1] \in \mathbb{N}$. $R_t$ is a normalised continuous value between 0 and 1, interpreted as the proportion of tweets at $t$ that classify as `Diagnosed`: 0 meaning all samples belong to `Control` and 1 meaning all samples belong to `Diagnosed`.

### 4.3 Results

*Figures 1* and *2* (see *Appendix A.4*) display the temporal mental health dynamics for the countries under investigation. The $R_t$ across different countries is a function of the country specific development dataset's distribution on which the classifier was trained. As such, the $R_t$ across countries are not directly comparable but are rather analysed by the momentum of how $R_t$ of a country changed over time and its divergence from $R_t$ of other countries.

### 4.4 Discussion

Foremost, we categorically cannot, nor do we, state that the temporal mental health dynamics are *caused* by the respective national lockdowns nor other measures, taken by governments to combat the spread of the virus. In this section, we offer interpretations in line with relationships discovered.

In the U.K. rate of depression ($R^{UK}$), we firstly observe the sharp, unsustained, increase of over 50% on Christmas day, before decreasing back to the status quo the next day. Upon further investigation we find that this phenomenon is well-documented (Hillard & Buckman 1982) and seeing that our classifier identified this phenomenon, without explicitly being aware of its existence, is encouraging. On March 9th, Italy National Lockdown begins onward we observe a sharp, sustained increase in $R^{UK}$ until March 23rd, U.K. National Lockdown begins, where $R^{UK}$ somewhat plateaus. We interpret this as an increase in anxiety amongst the U.K. population as neighbouring countries take decisive measures to slow the spread of the virus. A key theme in the build up to the U.K. national lockdown implementation was the intentional delay so that to ensure maximum utility from the policy[7]. However, a report published on the 16th of March

by the Imperial College COVID-19 response team[8] estimated that the the current combative approach taken up by the U.K. government would result in 250,000 deaths. The report was well-publicised by the British media and was arguably a factor in the pivot by the U.K. government.. This is somewhat supported by the change in $R^{UK}$ during the U.K. National Lockdown where we see a sustained *decrease* over the majority of the period.

The rates of depression of France ($R^{FR}$), Germany ($R^{DE}$), Italy ($R^{IT}$) and Spain ($R^{ES}$) behave differently from $R^{UK}$. Firstly, $R^{IT}$ increases sharply by over 100% in the initial days of the Italian National Lockdown. This can be interpreted as anxiety and concern as at this point Italy was regarded as the global epicentre of the pandemic. This was coupled with economic turmoil and great concern over the capacity of hospitals to handle the high requirements for intensive care units[9].

Similarly, a sharp increase in $R^{FR}$ over the initial days of the French National Lockdown period, after-which $R^{FR}$ rises throughout the lockdown period at a lower and inconsistent rate. A similar story could be tailored to $R^{ES}$. The major increase in $R^{DE}$ occurs in the preceding month, whilst during the German National Lockdown, $R^{DE}$ increases in the initial days, albeit at a lower rate. $R^{DE}$ then plateaus and decreases - creating a turning point in $R^{DE}$ during the German National Lockdown.

Furthermore, the $R$ of respective countries following the easing of respective lockdowns can be interpreted as the countries' outlook on the easing of restrictions. The French and Spanish populations experienced a reduction in symptoms of depression, such as anxiety, as is evidenced by the clear reduction in $R^{FR}$ and $R^{ES}$ respectively. We therefore conclude by, tentatively, stating that the easing of restrictions were received by an improvement in the mental state of the general populations of France

---

[7]ITV News. 2020. Coronavirus: Boris Johnson Announces UK Government's Plan To Tackle Virus Spread, Youtube. [online] Available at: https://www.youtube.com/watch?v=2U1YoKujYeY&list=PLFXSE3NhAYiZdb2qijJ7uemIB-IAYK5-y&index=893&t=0s [Accessed 1 September 2020].

[8]Ferguson, N., Laydon, D., Nedjati-Gilani, G., Imai, N., Ainslie, K., Baguelin, M., Bhatia, S., Boonyasiri, A., Cucunubá, Z., Cuomo-Dannenburg, G., Dighe, A., Dorigatti, I., Fu, H., Gaythorpe, K., Green, W., Hamlet, A., Hinsley, W., Okell, L., van Elseland, S., Thompson, H., Verity, R., Volz, E., Wang, H., Wang, Y., Walker, P., Walters, C., Winskill, C., Donnelly, C., Riley, Steven, R. and Ghani, A. 2020. Report 9: Impact Of Non-Pharmaceutical Interventions (Npis) To Reduce COVID-19 Mortality And Healthcare Demand. [online] Imperial.ac.uk. Available at: https://tinyurl.com/imperial-college-covid19 [Accessed 1 September 2020].

[9]CIDRAP 2020. Doctors: COVID-19 Pushing Italian ICUs Toward Collapse. [online] Available at: https://tinyurl.com/italians-covid19 [Accessed 8 August 2020].

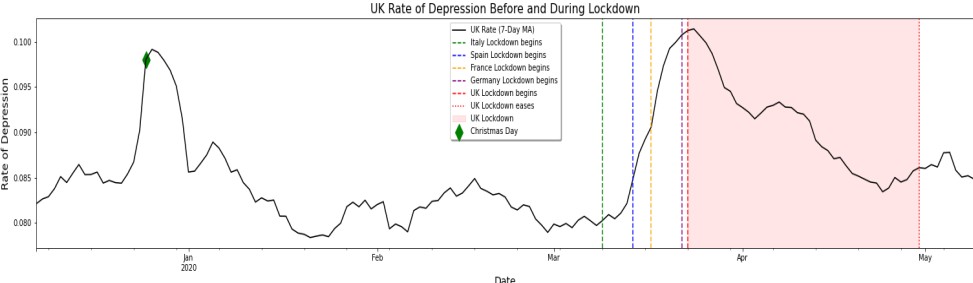

Figure 1: U.K. rate of depression before and during the National Lockdown. Noise in the rate of depression has been smoothed with a 7-day moving average.

and Spain, the mental state of the Italian and German general populations deteriorated, whilst the U.K. was agnostic to the easing of restriction.

We are hesitant to state the changes in $R_t$ had been *caused* by the imposition/easing of national lockdowns. To make such a claim we would be required to undertake a more fine-grained causality study which is beyond the scope of this paper, however we note this for future work. We *can* however claim to have discovered clear relationships between the drastic changes in the behaviour of rates of depression during the periods of the build-up to, during and in the aftermath of national lockdowns.

### 4.5 Ethical Principles

As we are proposing a public data driven approach for decision-making, we offer a discussion on ethics relating to possible exploitation of the system:

One such exploitation could arise where a pharmaceutical company, focused on the antidepressants market, utilising the methods proposed and analyse the rate of depression increasing in a particular country with no other access to an antidepressants supplier. The company could then proceed to monopolise the market and overcharge for their products thereby constraining the individuals financially, this may in turn increase levels of stress, anxiety and depression in the country. This would create an unethical reliance on the product arising directly from the implementation of the system.

Another, yet reversed, form of exploitation could arise by taking the scenario examined in the paper, if it was publicly known that the government of a country were to utilise the methods to decide how to proceed in the easing/re- implementation of the national lockdown, it is reasonable to assume that a third-party with a vested interest in the policy setting of the government could engage in activities to manipulate the publicly available data. This could come in the form of these individuals contributing high-volume data with the sole aim to skew and corrupt the data that will be mined and used for the decision-making of the governments.

This creates a trade-off dilemma between ethical principles currently within the social media platforms user agreements stating that users have the right to know how their data is being used with the need for partial secrecy in the exact mining methodologies and their end use which lacks transparency.

A prospective equilibrium to this trade-off would be the establishment of accountable Ethics Review Boards (ERBs) at the social media network companies that will be tasked with reviewing proposed systems, judged to be too sensitive to publicly expose, developments and implementations. Furthermore, these ERBs should be audited externally periodically to ensure of their integrity. High-level details of this proposed equilibrium should be added to the social media networks' user agreements to ensure that transparency, to the extent possible, is maintained.

## 5 Conclusion

Our set of experiments have been conducted with the aim of providing organisations with a methodology for monitoring and analysing temporal mental health dynamics using social media data. We examine sample representations and their ability to impact classifier performance. We investigate the role of an imbalanced dataset in the classifier training regime. Our classifier achieves encouraging performance on two fronts: the ability to discriminate, with reasonable performance, between `Diagnosed` and `Control` samples and identified the Christmas Depression phenomenon. Finally, we analyse the rates of depression and their relationships with respective national lockdowns.

## Ethics Board Statement

Having explicitly confirmed with the Queen Mary Ethics of Research Committee (QMERC) on a similar recent study which required Twitter data, we were advised that this type of research "does not need ethical review - being as it is analysis of data in the public domain". Additionally, we do not publicise any user IDs and/or text beyond the handful of expository examples which do not reveal any personal or identifying information. Finally, National Health Service (NHS) Research Ethics Committee (REC) review was not required for sites in England as per the online decision tool[10].

## Acknowledgments

Purver is partially supported by the EPSRC under grant EP/S033564/1, and by the European Union's Horizon 2020 program under grant agreements 769661 (SAAM, Supporting Active Ageing through Multimodal coaching) and 825153 (EMBEDDIA, Cross-Lingual Embeddings for Less-Represented Languages in European News Media). The results of this publication reflect only the authors' views and the Commission is not responsible for any use that may be made of the information it contains. We express our thanks to all of our data annotators: L. Achour, L. Del Zompo, N. Fiore, M. Hechler and R. Medivil Zamudio.

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

# A  Appendix

## A.1  Classifier Results on All Countries' Development Datasets

Table 7: $BILSTM$-$SELFA$ Classifier Performance on Countries' Development Datasets

| Country | Control | | | Diagnosed | | | Macro |
|---------|------|------|------|------|------|------|------|
| | P | R | F1 | P | R | F1 | F1 |
| France | 0.96 | 0.94 | 0.95 | 0.29 | 0.37 | 0.33 | 0.64 |
| Germany | 0.96 | 0.96 | 0.96 | 0.3 | 0.32 | 0.31 | 0.63 |
| Italy | 0.97 | 0.96 | 0.97 | 0.38 | 0.4 | 0.39 | 0.68 |
| Spain | 0.97 | 0.97 | 0.97 | 0.35 | 0.38 | 0.36 | 0.67 |
| U.K. | 0.94 | 0.94 | 0.94 | 0.32 | 0.31 | 0.32 | 0.63 |

## A.2  Significance of Results

Table 8: Significance in Predictions of $BILSTM$-$SELFA$ Classifier

| Country | Comparison | Significance |
|---------|-----------|--------------|
| France | Uniform | $\chi^2 = 1,311,459\ (p < 0.00001)$ |
| | Weighted | $\chi^2 = 6,726\ (p < 0.00001)$ |
| Germany | Uniform | $\chi^2 = 1,504,290\ (p < 0.00001)$ |
| | Weighted | $\chi^2 = 5,607\ (p < 0.00001)$ |
| Italy | Uniform | $\chi^2 = 1,485,204\ (p < 0.00001)$ |
| | Weighted | $\chi^2 = 5,050\ (p < 0.00001)$ |
| Spain | Uniform | $\chi^2 = 2,122,253\ (p < 0.00001)$ |
| | Weighted | $\chi^2 = 6,324\ (p < 0.00001)$ |
| U.K. | Uniform | $\chi^2 = 1,242,848\ (p < 0.00001)$ |
| | Weighted | $\chi^2 = 16,591\ (p < 0.00001)$ |

## A.3 Experiment Dataset Composition

Table 9: Composition of Experiment Datasets

| Country | Restrictions Begin | Restrictions Eased | No. Users | No. Tweets |
|---------|---------------------|---------------------|-----------|------------|
| France | 17 March 2020[1a] | 11 May 2020[2a] | $1,351$ | $945,919$ |
| Germany | 22 March 2020[3a] | 6 May 2020[4a] | $1,643$ | $998,248$ |
| Italy | 9 March 2020[5a] | 27 April 2020[6a] | $1,725$ | $764,089$ |
| Spain | 14 March 2020[7a] | 28 April 2020[8a] | $2,060$ | $1,012,847$ |
| U.K. | 23 March 2020[9a] | 30 April 2020[10a] | $2,883$ | $2,050,554$ |

[1a] The Independent. 2020. France Imposes 15-Day Lockdown And Mobilises 100,000 Police To Enforce Coronavirus Restrictions. [online] Available at: https://tinyurl.com/independent-covid19-france [Accessed 12 July 2020].

[2a] BBC News. 2020. France Eases Lockdown After Eight Weeks. [online] Available at: https://www.bbc.co.uk/news/world-europe-52615733 [Accessed 12 July 2020].

[3a] BBC News. 2020. Germany Bans Groups Of More Than Two To Curb Virus. [online] Available at: https://www.bbc.co.uk/news/world-europe-51999080 [Accessed 4 August 2020].

[4a] BBC News. 2020. Germany Says Football Can Resume And Shops Reopen. [online] Available at: https://www.bbc.co.uk/news/world-europe-52557718 [Accessed 4 August 2020].

[5a] CNN. 2020. All Of Italy Is In Lockdown As Coronavirus Cases Rise. [online] Available at: https://edition.cnn.com/2020/03/09/europe/coronavirus-italy-lockdown-intl/index.html [Accessed 12 July 2020].

[6a] BBC News. 2020. Coronavirus: Italy's PM Outlines Lockdown Easing Measures. [online] Available at: https://www.bbc.com/news/amp/world-europe-52435273 [Accessed 12 July 2020].

[7a] The Guardian. 2020. Spain Orders Nationwide Lockdown To Battle Coronavirus. [online] Available at: https://tinyurl.com/guardian-covid19-spain [Accessed 12 July 2020].

[8a] BBC News. 2020. Spain Plans Return To 'New Normal' By End Of June. [online] Available at: https://www.bbc.co.uk/news/world-europe-52459034 [Accessed 12 July 2020].

[9a] BBC News. 2020. Coronavirus Updates: 'You Must Stay At Home' UK Public Told - BBC News. [online] Available at: https://www.bbc.co.uk/news/live/world-52000039 [Accessed 12 July 2020].

[10a] BBC News. 2020. UK Past The Peak Of Coronavirus, Says PM. [online] Available at: https://www.bbc.co.uk/news/uk-52493500 [Accessed 12 July 2020].

## A.4 Temporal Mental Health Dynamics Results

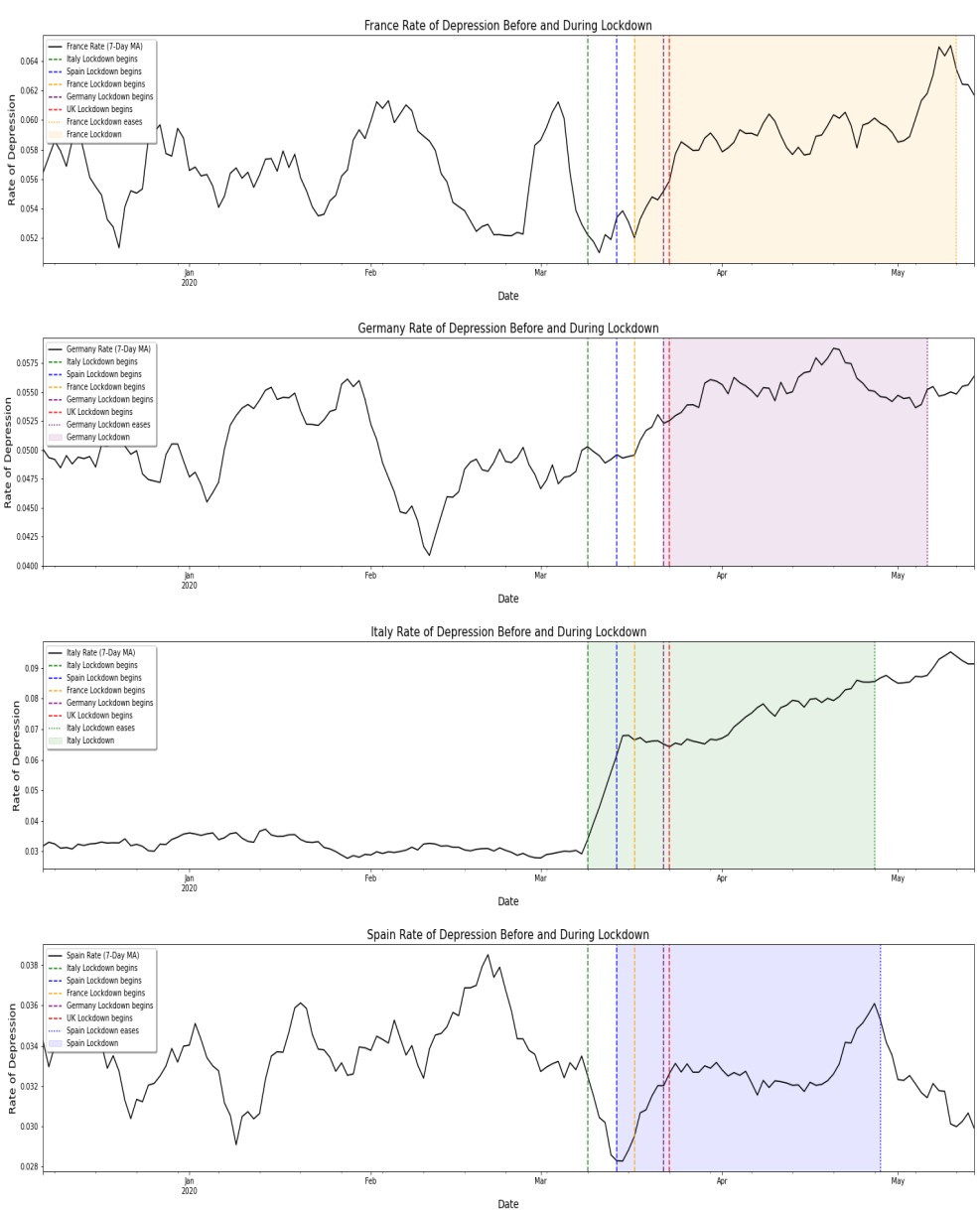

Figure 2: France, Germany, Italy and Spain rates of depression before and during respective national lockdowns. Noise in rates of depression have been smoothed with 7-day moving averages