# OpenReview forum: "Temporal Mental Health Dynamics on Social Media"
_EMNLP/2020/Workshop/NLP-COVID — NLP-COVID19-EMNLP Oral_

### Official Review · AnonReviewer3 · 2020-09-12
**Revise with new focus**

**Rating:** 4
**Confidence:** 4

**Review:**

### Recommendation
I recommend that this paper be revised and resubmitted as a short-paper, focusing on the unique challenges of multi-national mental-health predictions.

### Review
The paper aims to provide "organisations with a methodology for monitoring and analysing temporal mental health dynamics using social media data"; however, they do not gather enough data or train a sufficiently successful model to do that successfully. The paper does a good job of applying established methods for mental health analysis on social media data and the inclusion of a BERT classifier is compelling. The paper over-represents its usefulness in several areas.

The paper may be better suited as a 4-page short-paper. In that condensed format, the paper could focus on its unique contribution of analyzing data across national boundaries. This claim should be supplemented by country specific analysis of depression and suicidality. These are known to vary widely by culture.

Lastly, the authors should include a discussion of the ethical principals informing their decision making during this study.

Two additional nits: (1) the authors should include the AUC metric, which is conventional for this space; (2) the authors should use effect size or confidence intervals instead of significance testing, which has flaws on the massive N found in NLP.

### Summary

**Strengths**:
- Cross-national analysis
- Inclusion of a transformer-model
- Methodologically consistent with the literature

**Weaknesses**
- Small data set
- Poor performing models
- Lots of superficial information in the paper

### Reproducibility
This paper is moderately reproducible. The methods are conventional--well explained here and in previous works--however, the deep models are under documented and should be supported diagrams, more detailed description, or code. Those models, likely, could not be reproduced.

---

### Official Review · AnonReviewer2 · 2020-09-24
**A well written paper exploring the means for estimating population depression rates in multiple countries**

**Rating:** 6
**Confidence:** 3

**Review:**

In this paper, the authors attempt to develop a methodology to predict population depression rates across different countries from social media data (Twitter). The paper is well structured with clearly outlined aims, assumptions, and weaknesses. The discussion section provides a nice overview and is a good example of how machine learning results can be interpreted using domain knowledge. In my opinion, it indicates that the authors went the extra mile to gather information about the subject of their analysis.
The authors apply previously developed methods in a well-designed set of experiments, however, they manage to achieve only moderate performance.

Areas for improvement:
- The concept of a “distantly supervised dataset” should be clearly explained.
- Why were the classifiers described in section 3.2.2 trained for only 1 epoch? Training for longer might improve the performance of the classifiers.
- The authors also specify that Diagnosed samples were treated as 5 times more valuable. At what stage was this weighting implemented? How was number 5 chosen?
- How was the data split into training and validation sets?

Minor comments:
- The authors should specify the 2-week period within which the data described in section 3.1 was gathered.
- The false-negative example in Table 6 doesn’t sound like the person has been “diagnosed with depression”. Was it labelled as positive because of other, more affirming posts by the same user within a certain time period?
- In section 4, were tweets from different countries analysed by respective pre-trained classifiers?

---

### Official Review · AnonReviewer1 · 2020-09-26
**Construct distant supervised dataset for mining depression among tweets during COVID-19 pandemic.**

**Rating:** 6
**Confidence:** 3

**Review:**

The authors aim to identify the mental health dynamics on social media during COVID-19 pandemic. In this paper, they concentrate on identifying the people with depression through analyzing their tweets. They create a dataset by using a pre-existed distant-supervision methodology including cases (depression) and controls. They investigate the classification performance on SVM, and then turn to 8 classification models.

Strength：
The authors create a distantly supervised dataset and perform a careful investigation on the dataset, especially from the view of data imbalance.

Weakness:
 1. The algorithmic part is less novel nor informative, as the author mainly use the classical classification models.
 2. I am personally opt to regard the dataset created in this research as the main contribution. So, more accurate description of this part is expected. The dataset in this paper is created by a "distant-supervision methodology", but the details are not sufficient. More examples and the quality evaluation is expected.
 3. The authors could enunciate the performances comparison at section 3.2.2.

---

### Author Response · Authors · 2020-09-27
**Author Response**

Thank you all for your reviews. We would like to address your comments and question here:

Section 3.1:
- Further discussion of the distantly supervised dataset seems to be a reoccurring comment and is one that we have added in the camera-ready version. Furthermore, we are happy to share the datasets upon request.
- The 2-week period described in Section 3.1.1 is 1 July 2019 - 15 July 2019. We have added this detail in the camera-ready version.

Section 3.2.2.:
- An enunciation of the performance and comparison in Section 3.2.2. such as poor performance of BERT classifier will be further explored in the camera-ready version.
- Further details and considerations for reproducibility such as: sample weighting, validation split and single epoch training have been added and further discussed in the camera-ready version.
- Tweets from different countries were indeed analysed by respective pre-trained classifiers, Transfer Learning was not used in this process.

Additional comments:
- AnnonReviewer2's observation of the false-negative example is accurate, the sample was indeed tweeted by a user with a more affirming post regarding their mental health. This sample in particular highlights a limitation of the distant-supervision approach.
- An ethical principles section would make an interesting addition to the paper, not only in terms of the user data and sensitivity of the data gathered (mental health based) but also the deployment of a system that can be corrupted by those with a vested interest in the strategic decisions that are made based on system’s output.

We thank all the reviewers for spending their time providing us with insightful and much appreciated feedback. We are keen to reflect the proposed comments, most of which can easy be added, in the camera-ready version.